# Age and Maturation Matter in Youth Elite Soccer, but Depending on Competitive Level and Gender

**DOI:** 10.3390/ijerph20032015

**Published:** 2023-01-21

**Authors:** Honorato J. Ginés, Florentino Huertas, Tomás García Calvo, Jose Carlos Ponce-Bordón, António J. Figueiredo, Rafael Ballester

**Affiliations:** 1Doctorate School, Catholic University of Valencia “San Vicente Mártir”, 46008 Valencia, Spain; 2Faculty of Physical Education and Sport Sciences, Catholic University of Valencia “San Vicente Mártir”, 46900 Torrent, Spain; 3Faculty of Sport Sciences, University of Extremadura, 10003 Cáceres, Spain; 4Faculty of Sport Science and Physical Education, University of Coimbra, 3004-531 Coimbra, Portugal

**Keywords:** grassroot, maturity age, performance expectations, relative age effect, talent selection

## Abstract

This study aimed to explore the relevance of the relative age effect (RAE), maturity status and anthropometry, and their influence on coaches’ assessment of players’ performance, analyzing both genders and different types of academies (elite vs. non-elite). The sample included 603 soccer players (385 male), from the under 12 (U12), under 14 (U14) and under 16 (U16) categories, belonging to elite and nonelite teams. Coaches’ assessment of players’ performance, chronological age, anthropometric characteristics, maturity offset (MO) and peak height velocity (PHV) were registered. Our results showed that RAE was present in both genders within the elite, but not in the nonelite academies. Early maturity players were overrepresented in the male elite, but not in the female academies. No relationship was found between RAE and anthropometry in male elite academies. Male elite players showed better anthropometric characteristics than nonelite players, while this pattern of results was not found for female players. The coaches’ assessment on players’ current performance was not influenced by the chronological age nor anthropometry, but it was linked to the PHV. Coaches from nonelite academies rated better in current assessment of performance the taller players. Our findings suggest that maturity status and RAE play an independent and important role in the talent selection process.

## 1. Introduction

Soccer clubs spend increasing financial resources on talent identification (TI), selection and development processes [1]. These processes are very complex and diverse, as they involve many interacting factors in athletes’ performance [2]. Given the lack of objectivity in establishing the criteria for player selection and retention [3,4], clubs are professionalizing their own systems and structures [5], affording a greater number of opportunities to players with characteristics in line with the club’s identity [6].

Selection and TI processes are different depending on type of academies. Elite academies usually belong to professional or semiprofessional clubs [7]. In these academies players are selected through TI processes and training methodology is oriented to reach the highest sport performance and promote the best professional soccer players. Nonelite academies are usually linked to modest nonprofessional clubs with limited financial and human resources where the training process is mainly focused on having fun and developing educational values through sport practice [7].

Generally, the responsibility for the talent selection process during early stages falls on coaches and scouting departments, being reduced in many cases to observation, perception, pragmatism, and previous experience [8,9]. Technical ability, cognitive-perceptual skills, tactical skills [10], as well as anthropometric, physiological [11], and physical fitness factors [12], can influence the identification and selection of players, although all the process can be reduced to the final decision of the head coach or scouting department. However, subjectivity may involve less accurate decisions, missing out potential future talent [13].

One of the remarkable aspects within the TI process is the bias towards the selection of players with anthropometric advantages of being taller and heavier [14,15]. In fact, some studies have found anthropometric advantages when comparing elite and nonelite soccer male academies [16,17]. However, although previous research has shown that anthropometry can affect physical performance [18], other studies have reported non influence in overall soccer performance [19].

These anthropometric advantages are related to other potential bias in TI known as the relative age effect (RAE), occurring when subjects of different birth dates are grouped within the same chronological year of born to compete [20]. This presumption triggers physical and maturational (dis)advantages resulting from the interaction between birth date and grouping date [21]. This phenomenon is associated with a greater probability of selection and retention to be part of elite academies [22,23,24,25] in favor of those players born at the beginning of the selection cut [9,26]. These birth-date asymmetries may be determined by the more favorable evaluations of current performance of relatively older players [27]. Some studies have pointed that players born in the first quarter (BQ1) are three to four times more likely able to join elite academies than their BQ4 peers [28,29]. However, it has been shown that RAE tends to decrease in the later stages of soccer development [30]. Players from the last birth quarter (BQ4) who “survive” have higher probability of becoming professional players [31,32], being up to four times more likely to end up signing a professional contract than their BQ1 teammates [26].

Another important bias in the identification of sport talent is biological maturation, which although in early stages of development is more related to relative age of the players, should be considered independently as a different construct [27,33,34]. Biological maturation can be defined as the process of change that any young person undergoes, and that leads the athlete to complete his/her mature state in the different biological systems [35,36]. This process is individual and specific for each player, being asynchronous with chronological age [35,37]. Indeed, we may find players with a different maturational state within the same chronological age group. Some studies have shown maturational differences of 5–6 chronological years between subjects of the same age group [38,39], being early maturing players taller and heavier than their later maturing peers [12,36,40], and with physical advantages that can lead to greater motor and sports performance [41,42,43].

Physical, anthropometric and maturity status biases may imply a substantial loss of potential talent of relatively young or late-maturing players [44], due to, among others, the importance attributed to obtain the highest short term or current performance over the intermediate goals for supporting the basis of future player development [45]. In fact, some studies point to higher prevalence of relatively older players when comparing elite vs. nonelite academies [46,47]. These differences have also been shown in the female gender [48]. However, it is necessary to take into account that the aforementioned studies were mainly descriptive approaches that did not consider the anthropometric characteristics and the maturational status of the players.

In this context, it is also interesting to build on this previous research to further explore the potential relationship between anthropometrics and players’ performance assessment by their coaches, since the expectations placed by them highly influence the dropout rates and the development of their sporting career [49,50,51].

For these reasons, our research is highly novel since analyzing, from a gender perspective, the differences in the manifestation of RAE and other athlete selection biases as a function of the type of academy (elite vs. nonelite) in soccer. The objectives of the present study are twofold: (a) to analyze and compare the birth distribution, maturity status, and anthropometric characteristics of male and female players from elite and nonelite academies; (b) to explore the potential relationship between the coaches’ assessment of player´s present and future performance, with the chronological age (or RAE), maturity status, and anthropometric characteristics of the players.

Due to the higher level of professionalism in men’s football, the greater number of players, and the economic and human resources available for the recruitment and promotion of talent, larger chronological age and maturation biases were expected in men’s football. Similarly, we predicted a greater magnitude of the effect of the RAE and maturational biases in the selection process (taller and more mature players) in the elite compared with the nonelite academies. Accordingly, we also expected, especially in early stages, an influence of the maturational variables in the assessment of current performance and the expectations of the future potential of young players.

## 2. Materials and Methods

### 2.1. Sample

The present study was carried out within different soccer academies (elite vs. nonelite) in the Valencian community (Spain). A total of 603 players, 385 male (145 elite, 240 nonelite) and 218 female (127 elite, 91 nonelite) where divided according to their chronological age (U12, U14, U16). As elite were considered the academies of professional clubs within the Spanish First Division League in the 2021–2022 season (LaLiga Santander), while the rest of the academies of clubs competing lower than in the fourth division of Spain (2nd RFEF), were defined as nonelite.

Parents and players signed agreements for the use of their data for internal and external club´s research purposes. In any case, parent-mentors and players signed a consent and assent document including information about the date of birth, team, and category. Present research received the approval by the University’s Research Ethics Committee (UCV/2019-2020/149), in accordance with the ethical guidelines of the Declaration of Helsinki.

### 2.2. Data collection and Procedures

#### 2.2.1. Chronological Age and Relative Age Effect

Currently, the youth academies leagues within the Spanish Soccer Federation are established considering the chronological year of the players. Responsible of the academies from each clubs provided information about the date of birth of each player, being grouped according to their date of birth in one of the four quarters that make up the chronological year (1 January to 31 December): BQ1 (1 January to 31 March), BQ2 (1 April to 30 June), BQ3 (1 July to 30 September), and BQ4 (1 October to 31 December).

#### 2.2.2. Anthropometric Characteristics

The players were cited 30 min before the beginning of training season. Anthropometric characteristics were collected in standardized conditions (16 ± 2 °C) inside the sports facilities from each club.

Standing and sitting height were measured using a Seca 206 tape (“Physical distancing for health”, Hamburg, Germany), with a 0.1 cm accuracy using the ISAK protocol (International Society for the advancement of the Kinanthropometry). Body weight was determined using Tanita SC 240 MA scales (±0.1 kg), and with the players wearing their training clothes (socks, T-shirt, and shorts) [52].

Two measurements were taken for standing and sitting height and weight. When height and body mass measures differed by more than 0.4 cm and 0.4 kg, respectively, a third measurement was taken, and the mean value was assigned [53].

#### 2.2.3. Maturity Status

The MO (maturity offset) was estimated through a noninvasive method, appropriate for the age range of our sample, using a series of anthropometric measurements: standing height (cm), sitting height (cm), lower limb height (cm), body mass (Kg), and chronological age (CA) [54]. The Mirwald formula was carried out [55].

The peak height velocity (PHV) period, which indicates the theoretical reference point of maximum height growth, being the most commonly used indicator to determine somatic maturation [56], was estimated by subtracting MO from the chronological age PHV = CA−MO [55,57]. Previous research showed that estimation of the PHV age systematically increases with chronological age [56], and that depending on the context, the PHV age is approximately between 13.3 and 14.4 years for boys and between 11.3 and 12.2 for girls [35,58], and with higher growth curves for height and body mass [35,53]. In our research the estimation of the maturity status was fixed through identifying the PHV of each age group analyzed and its specific standard deviation (ST) were used to construct the groups [36,59] (Table 1). Once the PHV is established, players can be classified as early, on time or later (see Appendix A to check the calculations of this variables).

#### 2.2.4. Coaches’ Assessment of Current and Future Players’ Success

To evaluate coaches’ assessment of current and future players’ success, direct questions were asked about the level of performance nowadays (current performance) in their team, and what the future potential to become an important player was. These direct questions are appropriate for certain circumstances, especially to measure at the individual or collective level, on issues of efficacy or performance [60,61]. Each question included a Likert scale (1 to 5), and each player was rated according to the average level of their team, where the comparison was between their own teammates from Level 1 (far below average) to 5 (far above average). The questions were: (a) “Please, quantify the individual current player´s performance, taking into account all aspects and demands that influence the game (technical-tactical, physical, psychological-emotional, etc.)”; and (b) “Please, quantify the individual future potential performance of the player, when he/she has reached sporting maturity, taking into account all aspects and demands that influence the game (technical-tactical, physical, psychological-emotional, etc.)”.

Questionnaire was provided to all the members of the coaching staff (elite teams: mean of 3.23 members per team; nonelite teams: average of 1.87 members per team) via link to the Google Forms web survey platform.

#### 2.2.5. Statistical Analysis

The present study used the statistical package IBM SPSS Statistics Statistical (Version 25.0, Chicago, IL, USA). A descriptive analysis of the distribution of birth dates, grouped by trimester, was carried out for each age group, gender, and type of academy.

Frequency counting was used to determine the number of players within each birth quarter (BQ1, BQ2, BQ3, BQ4) and maturity status (early, on time, late). Chi-square goodness-of-fit tests were used to test for homogeneity on the one hand; in the distribution of the established groups according to date of birth, and the other hand according to the maturity status, to obtain in both the analysis of differences in expected and observed frequency. In the case of BQ, the theoretical assumption of uniform births during the different quarters of the year (25% per quarter) was adopted [62], this distribution is found in most countries, where no variations between BQ are shown [63]. In the case of the maturity status, a normal distribution of the sample is assumed, with a high percentage of players from on time, and a lower and similar percentage from early and late groups.

As such, the chi-square test does not reveal the magnitude and direction of an existing relationship; when statistically significant differences were found, odds ratios (OR) and 95% CIS for quarters (BQ1 vs. BQ4) and semesters (S1 vs. S2) were conducted. An OR of 1.00 indicated that the probability of belonging to one group or the other is equal, while 2.00 OR denoted that the probability of belonging to one group or the other is double [21].

Normal distribution and homogeneity of variance were checked (*ps* < 0.05), by the Kolmogorov-Smirnov and Levenne’s tests respectively in all our dependent variables (weight, height, sitting height, decimal age, present performance, potential future performance).

Differences in assessment of player´s performance by trimester of birth and maturity status were analyzed by one-way analysis of variance (ANOVA), with post-hoc test (Bonferroni).

For the comparison between two groups (elite vs. non-elite academies or male vs. female), Student’s t-test for independent samples were used.

The relationship between anthropometric measures (weight and height) and assessment of player´s performance (current and future expectations) was analyzed using Pearson’s correlation coefficient, adopting a significance level of *p* < 0.000. The level of significance for the rest of statistical procedures was set at (*p* = 0.05).

## 3. Results

### 3.1. Chronological Age and Relative Age Effect (RAE)

Considering the whole sample, our results showed that the RAE is present in elite academies X2 (gl3, N = 271) = 68.498; *p* = 0.000, OR = 4.72; *p* = 0.000, while this phenomenon does not occur in nonelite academies (OR = 0.90; *p* = 0.642).

Analysis by gender showed that RAE was observed only in the male elite academies (*p* < 0.05) (Figure 1), where BQ1 presents 53% of players compared to 4% of BQ4. RAE also appeared significantly in all the age categories (*p* < 0.05), increasing its size as the age category is older, highlighting the absence of BQ4 players in the U16 category (Table 2).

However, in the nonelite academies, the RAE was not present (*p* > 0.05), neither in the total sample analyzed (26% BQ1 and 26% in BQ4,) nor in any of the age categories analyzed (*p* > 0.05) (Table 3), where U14 and U16 showed a higher number of players born in the second semester.

Considering women academies, the RAE was also present within the elite academies when considering the total sample (*p* < 0.05), with 32% BQ1 compared to 15% BQ4 (Figure 2). However, RAE was not found in any of the analyzed age categories U12, U14, and U16, although all categories showed a higher number of births in the first semester of the year (Table 2).

RAE was not present in nonelite academies when considering the total sample (*p* > 0.05), with 22% BQ1 vs. 33% BQ4 (Figure 1), and in neither of the age categories (*ps* > 0.05), founding a similar distribution of U12 players born in both semesters, and a greater number of players born in second semester in the U14 and U16 groups (Table 2).

### 3.2. Maturity Status

Due to the categorization of the player´s stage of maturation, a normal distribution of the total sample was observed, with a higher percentage for players on time 75.1% male, and 66.8% female. In both genders from the elite academies an overrepresentation of early mature players and a decrease in late mature players were found according to the specific moments of maturity, but with much more marked differences in the males (Table 4).

Weight (heavier), height (taller), and PHV (arrives earlier) showed significant differences (*ps* < 0.05) in athletes classified as “earlier” compared to “later” maturity status, in all age categories, academies, and genders.

### 3.3. Relationship between RAE and Maturity

Figure 2 show the distribution of the maturity status according to the player´s birth quarter. In males, the percentage of early maturity players was higher in all the BQs in elite than nonelite academies. This pattern was totally contradictory when considering nonelite academies, showing a higher percentage of late maturity players than in elite academies (note that in elite academies there were not late maturing players from BQ4).

Considering female gender, the elite academies presented a greater number of early maturing players in the last BQs than not elite (BQ3 = 21.4% vs. 13.0%; BQ4 = 27.8% vs. 16.70%, respectively). However, elite academies showed a higher percentage of late maturity players in BQ2 (15.4% vs. 5.60%) and BQ3 (17.90% vs. 13.0%) than their nonelite counterparts.

### 3.4. Anthropometric and RAE

#### 3.4.1. Anthropometric and RAE within Elite and Nonelite Academies

ANOVAs were carried out to compare the changes in anthropometric variables considering the BQ, gender, and the type of academy to which the player belongs.

In the analysis of the elite male academies, in the U12 group, players belonging to BQ1 and BQ4 were taller than their BQ2 and BQ3 teammates (*p* < 0.05), while within the U14 and U16 groups, no differences were observed (*p* > 0.05). Within the nonelite academies, U12 players born in BQ1 were taller than their BQ3 (*p* = 0.006) and BQ4 (*p* = 0.009) counterparts. Regarding weight, significant differences were found when comparing players born in S1 with the S2 peers in the U12 category, showing that the older ones were heavier that those born in the first half of the year (t = 2.471; *p* = 0.016) (Table 3).

In the female gender, the quarter of birth did not modulate the player´s weight or height in any of the two types of academies analyzed (*ps* > 0.05) (Table 3).

#### 3.4.2. Anthropometric Comparison between Elite and Nonelite Academies

Table 5 shows a descriptive analysis of the anthropometric variables (weight and height) within each age group, considering gender and type of academy.

Male soccer players from elite academies showed significant advantages (older and taller) (*ps* < 0.05) in all the age categories with respect to their nonelite counterparts. However, analyses of weight and BMI showed significant differences in weight in only U16, displaying that players from elite academies were heavier than those from the nonelite (*p* < 0.05), and in BMI in U12 in favor of nonelite players (*p* < 0.05).

Considering female players, only a slight tendency to an older chronological age was found for U16 elite academy players (*p* = 0.085). No differences have been observed in other anthropometric measures, except in height, revealing that players from elite academies were taller than nonelite in the U14 and U16 categories (*p* < 0.05). It is noteworthy that the U12 category shows parameters very similar in all the variables analyzed.

#### 3.4.3. Anthropometrics and Coaches’ Assessment of Players’ Performance

Correlational analyses were run to explore the association between anthropometrics (height and weight) with the coaches’ assessment of players’ performance, both currently and in the future (Table 6).

Within elite male academies, no associations were found between player’s height or weight and coaches’ assessments. However, in men’s nonelite academies, coaches from U12 and U16 teams linked the current performance and height (r = 0.375; *p* = 0.001; and r = 0.334; *p* = 0.002, respectively). Moreover, coaches from these academies associated player height with their future expectations of success in U12 (r = 0.247; *p* = 0.037), and in U16 (r = 0.355; *p* = 0.001), not obtaining any connection with respect to weight.

In the case of women’s soccer, no connection was found between anthropometric characteristics and coaches’ assessment within the elite academies, while in the nonelite academies, only the coaches of U16 associated the taller players with better current player performance (r = 0.363; *p* = 0.035).

#### 3.4.4. RAE and Coaches’ Assessment of Player’s Performance

Coaches did not link their assessment on player’s success to the age cutoff point in general (Table 7).

When analyzing the male gender, coaches from elite academies did not associate their assessment of athlete’s current performance nor future expectations of success to the RAE (*ps* > 0.05), only in the U16 group in current performance (*p* < 0.05) in favor of BQ2 (M = 3.48 ± 0.8) and BQ3 (M = 3.47 ± 0.8) when compared to BQ1 (M = 2.94 ± 0.7), this age group had not got any player from BQ4. It should be noted that players from BQ4 were scored lower on their coaches’ future expectations. Regarding nonelite academies, no significant relationships were observed between these variables.

Considering female elite academies, significant differences were found in the U12 and U14 categories when analyzing coaches’ future expectation and RAE (*p* < 0.05). In the U12 group, the BQ3 players obtained better expectations than players from BQ4 (*p* = 0.021), and a trend of BQ1 compared to BQ4 (*p* = 0.065). In the U14 group, the BQ2 players received better future expectations than BQ3 (*p* = 0.019). No significant differences or trends on these associations were observed in coaches from nonelite academies.

#### 3.4.5. Maturity and Coaches’ Assessment of Player Performance

Our ANOVAs analyzed all elite and nonelite categories across both genders. Results showed that only nonelite male U12 academy coaches associated the player’s future performance with maturation (F = 3.616; *p* = 0.032), scoring early maturity players better (M = 4.33 ± 0.82) than the late maturity counterparts (M = 3.50 ± 0.61). No significant differences (*p* > 0.05) were found in the rest of the age categories.

To analyze the male elite U16 category, a T-student analysis was used, since the late-maturing group did not show any players. Results showed better expectations for early maturing players compared to on-time, both in current performance (t = 2.079; *p* = 0.042) (M = 3.53 ± 0.66 vs. M = 3.04 ± 0.82), as in future performance (*t* = 2.292; *p* = 0.026) (M = 3.53 ± 0.66 vs. M = 3.04 ± 0.82).

Regarding the relationship between PHV and the assessment of player performance in the male gender, only an inverse relationship in the male elite U16 group was found in expectations of future performance (*r* = −0.279; *p* = 0.034), showing that the earlier the players reached the PHV, the better expectations for success were received. Similar trends were also observed in nonelite U12s on current performance (*r* = −0.223; *p* = 0.060), and future expectations (*r* = −0.214; *p* = 0.072), favoring in both those players who reached the PHV earlier (Table 6).

No relationship between maturity status and coaches’ assessment of player’s performance was found in the female gender in any age category, nor type of academy analyzed *(ps* > 0.05).

## 4. Discussion

The present study explored the distribution of RAE and maturity status in soccer players of different age groups (U12, U14, U16), discerning between male and female gender, and considering the academies competitive level (elite and nonelite). Coaches’ assessment of the players’ current performance and expectations for future success were also considered to explore their relationships with player’s anthropometrics, chronological age, and maturity status. Our findings are partially consistent with those obtained in previous studies developed in different countries showing that RAE is present in male and female elite academies, without appearing in the nonelite environments [44,46,64]. Our results showed that the probability of belonging to BQ1 is much higher than belonging to BQ4 in both genders: male (OR = 12.83) and female (OR = 2.16), and when comparing by semester of birth: male (OR = 3.39) and female (OR = 1.70). Therefore, and agreeing with other studies, soccer shows a much more prevalent RAE in males than in females, even at the grassroot stages [65,66].

Regarding the age categories, in the male elite academies, birth asymmetries were found in all the age groups studied in our investigation. Interestingly, the overrepresentation of players born in BQ1 as compared to BQ4 increased with age: U12 (OR = 5.25), U14 (OR = 11.5), and U16 (no players born in BQ4). These results are coherent with those obtained from an English soccer academy [67]. This pattern of results have not been shown in the female elite teams, probably due to a less competitive environment, the wider range of player’s ages in the same team, and the lower level of the available resources in the scouting and talent detection departments compared with the male academies.

Some contradictory results have appeared in the literature on how gender or age modulate the RAE. Some studies reported that the RAE tends to decrease as one moves into an older age category within elite academies [30,47]. This controversy in the distribution of the RAE by age category could be partially explained by the context and philosophy of the club. In the elite academies, our findings have confirmed that talent selection process seems to be biased to favor those players born in the first quarters of the year, probably showing early maturation, and providing them better opportunities to develop their talent and achieve the success within a professional environment [26,68]. RAE in women soccer in Spain and other countries is increasing in the last years due in part to the growth and professionalization of female football, as can be seen when comparing different studies in recent years [48,65,69]. Some countries have tried to reduce the RAE in the selection process, but it does not seem to have had an effect, and there is still a much higher probability of selection for those born in the first quarters of the year [70].

Our results have confirmed that RAE is a physical bias still present in the selection of players in the Spanish elite academies. However, and according to preceding research, the study of physical biases is incomplete if it is not also considering the maturity status of the players [27,33]. Our findings showed that maturity status of our sample follows a normal distribution, with an overrepresentation of players on time (>70%), and lower and similar percentages (around 15%) in early and late maturity players. These findings are similar to those previously reported by an English soccer academy showing 84.8% on-time players, while for early 9.5%, and late 5.7% [34]. As expected, players classified as “earlier” were heavier and taller than their “later” counterparts, in all the analyzed age categories, academies, and genders. More interestingly, our results showed that, while in female gender the PHV period was similar in both types of academies, in males, players from elite male academies arrived at the PHV before their nonelite peers (*p* < 0.05). There is controversy about whether this is something common in soccer academies, since some studies have reported a similar pattern of results [71], while others did not find these differences in the age of the PHV in players from U11 to U21 in the United Kingdom (UK) [31]. In our study, we found a larger number of early maturing players in elite academies (20.7% vs. 7.1%), and lower number of late maturing players (6.2% vs. 16.7%). These data demonstrate a greater probability (OR = 3.33) of entering in elite academies if one is an early mature player. These results are coherent with those obtained from Portuguese U14 categories, reporting greater number of early elite mature players and late matures nonelite players [72]. However, no statistically significant differences between elite and nonelite maturation distribution were found in females, although a trend to replicate the findings in males was observed, and therefore, this relationship should be explored in future research using bigger sample sizes in different types of academies.

Despite the importance of analyzing RAE and maturation independently, here we also considered it highly interesting to investigate the relationship between both factors. Remarkably, our results have shown that in elite male academies, 33% of the players born in the second semester matured early, and barely 2% matured late. These data are coherent with our findings showing that the PHV age comes significantly earlier in players from elite than nonelite academies, for players born in the second half of the year compared to players born in the first one, reaching the PHV 0.5 years earlier in the male gender between BQ4 and BQ1, and 0.4 for females. These results are similar to others described in English academies showing differences of 0.3 years in the earlier age categories of U9, disappearing as one advances through the following age categories [67]. Despite our results by age categories showing more pronounced differences in U14 and U16 ages, our findings show that players from elite academies reach maturity before their chronologically older peers. However, inferences obtained from the PHV data should be considered with caution due to the limitations of the PHV estimation processes, wherefore the moment, date, place, race, measurement protocol, etc., can alter the results or the comparison between participants from different studies.

According to the previous literature, anthropometric development seems to underlie the physical biases we have described above (maturity status and RAE) [16,40,73]. Indeed, our results showed a direct relationship between maturation, relative age, and anthropometrics. Chronological data confirmed our maturity results, showing that male players born in BQ1 were taller and heavier than their peers born in BQ4, being significant (*p* < 0.05) in U12 and U14, but not in U16. This pattern was similar in female players, although these differences were only observed in U16 (*p* < 0.05), and a trend in U14 (*p* = 0.072), probably due to the sample size and variability in the characteristics of the academies. These results are coherent with those obtained from an UK elite club showing no differences in any category from U11 to U18 [67] but contradict another study in Belgium reporting anthropometric differences influenced by the RAE (BQ1 vs. BQ4), especially in younger categories, U13 and U15, but not in U17 [43]. These controversial findings can be partially explained by differences in the individual maturity status, since the anthropometric differences tend to be reduced when the athletes approach PHV age or beyond [55].

The relevance of RAE and maturity on talent detection is confirmed by the fact that players from elite academies were taller than their nonelite peers (*p* < 0.05). A similar trend was found in other studies examining Portuguese players from 11 to 15 years of age [16] or U11 to U21 players in the UK [31]. These results could be explained by the fact that the differences in the demands of a competitive context can play a homogenizing role in the players at elite academies, providing an important role for anthropometry in the selection process.

On the other hand, physically disadvantaged players who manage to progress in their careers within elite academies show much higher technical and skill levels than their peers, due in part to the different adaptations they have had to make to facilitate their physical disadvantages; this phenomena is explained within the umbrella of the “underdog hypothesis” [26]. Trying to shed light on the selection and development process, one of the purposes of our study was to explore whether coaches’ assessment of present and future players’ performance might be influenced by physical biases, as the beliefs of the coaching staff are fundamental in the retention and attention devoted to their players [31,74]. The coach’s subjective opinion is something common in talent identification and development processes around the world [75,76].

Our results showed that RAE does not seem to influence coaches’ performance assessment in nonelite academies but also in male players’ elite academies. These results replicate those obtained in a German talent training program [77]. However, our results showed a pattern of RAE in the elite female academies, showing that the chronological age is related to the future expectations for success in U12 and U14 players (players born in BQ4; U12 and BQ3; U14 were worse valued). Conversely, the maturity status was correlated with coaches’ performance assessment, especially of the male elite players; early mature players obtained a better assessment of current a future performance. This result is highly interesting because, while coaches are probably not conscious of this maturational bias in their performance assessment, this fact may imply a more favorable treatment in the selection and development process of early maturing players [20,78]. However, this results should be interpreted with caution as we have not found any relationship between anthropometrics and coaches’ performance assessment in elite academies, in line with previous studies using qualitative interviews [79]. This is something that could be critical in academies looking for short term performance. In our study we found that in nonelite academies, higher current expectations were assigned to the taller players in the U12, U16 male, and U16 female teams. However, this relationship between anthropometrics and expectations was not observed in other previous study with a lower sample size with U13 and U15 first and second level academies [51].

### 4.1. Limitations and Future Directions

This study increases the knowledge about this research topic; however, a few limitations could be identified with a view to further research.

It must be considered that the coaches rated their players establishing comparisons with their teammates, so the sample may already be biased in the selection process, that is really strict in the elite academies analyzed. It should be also taken into account that the coaches evaluations are subjective in nature, which makes it, despite the premises established in the protocol, more difficult to establish comparatives between teams [80], as each coach has his/her own background, training experience, game philosophy, etc.

More extensive research is required to keep exploring talent identification and development. Future research should consider how physical biases may influence physical but also technical-tactical performance. Taking this into consideration, the exponential growth of women’s football opens up an interesting venue to keep exploring relevant factors influencing talent identification and development in a flourishing context.

### 4.2. Conclusions

In the present study, an overrepresentation of players born during the beginning of the year was found in both male and female elite academies, a phenomenon that was not manifested in the nonelite academies, which presented a similar distribution in the birth quarters.

Our results reinforce, adding new evidence in female players, the findings from previous studies about the influence of chronological age and maturity status on talent selection processes. Anthropometrics also seem to be an important factor in the talent selection processes, depicting our sample differences in favor of elite players when comparing with players from nonelite academies.

Physical biases (RAE, maturity, and anthropometrics) influenced coaches’ performance assessment. Interestingly, the prevalence and magnitude of the physical biases on performance assessment were modulated by the competitive level and the age of the players. Therefore, it is important for the coach to be aware of the potential physical biases when evaluating and placing expectations on players of different ages, as it could have an influence on players’ short-term performance and development.

### 4.3. Practical Applications

The results obtained in our study are useful to help raise awareness of relative age and maturational biases among professionals involved in the processes of identifying and developing sporting talent [33]. We have added some Appendix A to our paper to help the responsible of sport training to calculate the player´s maturation. We believe that the inclusion of individual maturational status assessment in longitudinal player assessment is simple, quick, and inexpensive. Given that the state of maturity will vary throughout the player development process, we consider crucial to adjust the developmental objectives according to the individual maturational state of the young player. This will optimize the individual effect of the training loads and reduce injury rates.

On the other hand, as performance assessment may be influenced by physical biases, we suggest that the subjective opinion of the coaches and scouts on players’ potential should be complemented with objective data on their maturational state provided by a sport scientist (i.e., variables described in our investigation related with relative age, anthropometry, and maturity status). This will add more context to the coach and scouts view and will prevent the effects of physical biases in young players’ development.

Therefore, continuous control of players’ maturational state will allow clubs and coaches to carry out adaptions in preventive work and warm-ups, reducing injury risk. Clubs can even go a step further and carry out exercises and matches where players are grouped by their biological age instead of their chronological age to promote players’ development in a context of more equal opportunities for all.

## Figures and Tables

**Figure 1 ijerph-20-02015-f001:**
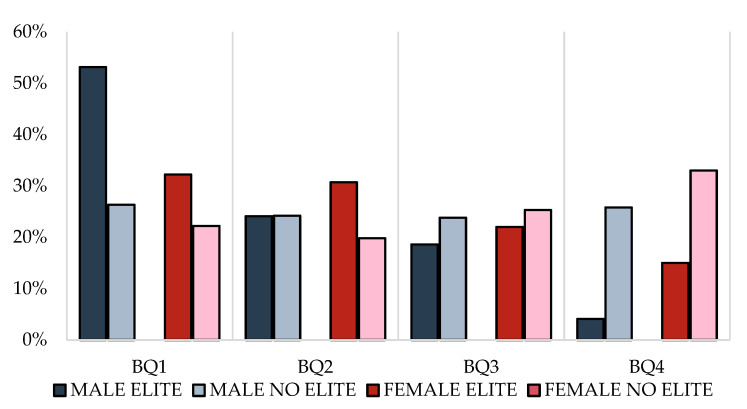
Relative age effect in male teams: BQ1—first quartile; BQ2—second quartile; BQ3—third quartile; BQ4—fourth quartile.

**Figure 2 ijerph-20-02015-f002:**
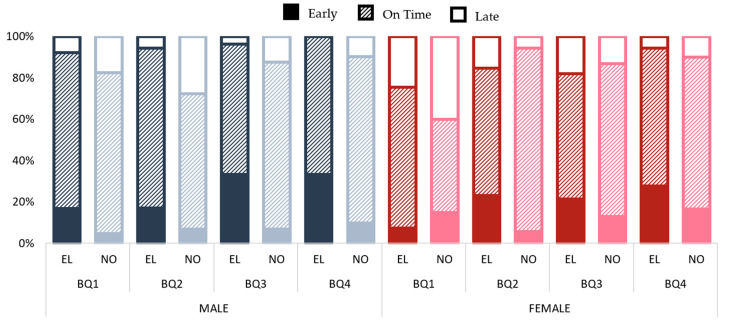
Maturity status distribution by birth quarter in elite and nonelite in male and female academies. EL = elite; NO = nonelite.

**Table 1 ijerph-20-02015-t001:** Classification of the maturity status according to the PHV of each age group.

Gender	Category	APHV	Maturity Status		
			Early	On Time	Late
Male	U12	13.44 ± 0.503	<12.94	12.94 to 13.94	>13.94
	U14	13.91 ± 0.553	<13.35	13.35 to 14.46	>14.46
	U16	14.15 ± 0.871	<13.28	13.28 to 15.02	>15.02
Female	U12	11.91 ± 0.505	<11.40	11.40 to 12.41	>12.41
	U14	12.17 ± 0.389	<11.78	11.78 to 12.56	>12.56
	U16	12.79 ± 0.498	<12.3	12.30 to 13.29	>13.29

Note: APHV = age of peak height velocity.

**Table 2 ijerph-20-02015-t002:** Relative age effect in the different elite and nonelite academies, differentiating between genders.

Gender	Category	Academy	Quarter								Total	Chi-Square			OR (95% CI)	
			BQ1		BQ2		BQ3		BQ4							
			n	%	n	%	n	%	n	%		X^2^	gl	*p*	BQ1/BQ4	S1/S2
MALE	U12	Elite	21	43.8	10	20.8	13	27.1	4	8.3	48	12.500	*3*	**0.006**	5.25 (1.38–19.9)	1.82 (0.80–4.13)
		Nonelite	20	27.8	20	27.8	15	20.8	17	23.6	72	1.000	*3*	0.801	1.17 (0.47–2.95)	1.25 (0.65–2.41)
	U14	Elite	23	59.0	10	25.6	4	10.3	2	5.1	39	27.564	*3*	**0.000**	11.50 (1.93–68.5)	5.50 (1.89–16.0)
		Nonelite	16	19.3	25	30.1	21	25.3	21	25.3	83	1.964	*3*	0.580	0.76 (0.32–1.82)	0.98 (0.53–1.79)
	U16	Elite	33	56.9	15	25.9	10	17.2	0	17.2	58	15.138	*3*	**0.001**	-----------	4.80 (2.04–11-3)
		Nonelite	27	31.8	13	15.3	21	24.7	24	24.7	85	5.118	*3*	0.163	1.12 (0.47–2.68)	0.89 (0.49–1.62)
MALE	TOTAL	Elite	77	53.1	35	24.1	27	18.6	6	4.1	145	73.455	*3*	**0.000**	12.83 (4.87–33.8)	3.39 (2.05–5.63)
		Nonelite	63	26.3	58	24.2	57	23.8	62	25.8	240	0.433	*4*	0.933	1.01 (0.61–1.68)	1.01 (0.71–1.45)
FEMALE	U12	Elite	16	38.1	10	23.8	10	23.8	6	14.3	43	4.857	*3*	0.183	2.67 (0.74–9.63)	1.62 (0.68–3.87)
		Nonelite	5	25	5	25.0	6	30.0	4	20.0	20	0.400	*3*	0.940	1.25 (0.22–7.08)	1.00 (0.29–3.45)
	U14	Elite	11	24.4	18	40.0	9	20.0	7	15.6	45	6.111	*3*	0.106	1.57 (0.47–5.23)	1.81 (0.78–4.20)
		Nonelite	6	16.2	7	18.9	9	24.3	15	40.5	37	5.270	*3*	0.153	0.40 (0.10–1.56)	0.54 (0.20–1.48)
	U16	Elite	14	35.9	11	28.2	9	23.1	5	12.8	39	4.385	*3*	0.223	2.80 (0.72–10.7)	1.79 (0.72–4.40)
		Nonelite	9	26.5	6	17.6	8	23.5	11	32.4	34	1.529	*3*	0.675	0.81 (0.21–3.22)	0.79 (0.30–2.05)
FEMALE	TOTAL	Elite	41	32.2	39	30.7	28	22.0	19	15.0	127	9.913	*3*	**0.019**	2.16 (1.05–4.44)	1.70 (1.03–2.81)
		Nonelite	20	22.0	18	19.8	23	25.3	30	33.0	91	3.637	*3*	0.303	0.67 (0.29–1.53)	0.72 (0.53–1.28)
TOTAL		Elite	118	43.4	74	27.2	55	20.2	25	9.2	272	66.971	*3*	**0.000**	4.72 (2.75–8.11)	2.40 (1.69–3.42)
		Nonelite	83	25.1	76	23.0	80	24.2	92	27.8	331	1.677	*3*	0.642	0.90 (0.59–1.37)	0.92 (0.79–1.08)

Note: BQ1 = first quartile; BQ2 = second quartile; BQ3 = third quartile; BQ4 = fourth quartile; n = number; % = percentage; x^2^ = chi-square value; gl = freedom degree; *p* = significance value; Or = odds ratio and 95% confidence intervals (95% CI); BQ1–BQ4 = first versus last quartile; S1/S2 = first versus last half year’s distribution.

**Table 3 ijerph-20-02015-t003:** Anthropometric and RAE in all categories.

Gender	Category	Academy		Quarter				GL	F	*p*
				BQ1	BQ2	BQ3	BQ4			
Male	U12	Elite	Height	153.8 ± 9.8	146.7 ± 4.7	146.7 ± 4.8	150.2 ± 0.6	3	3.366	**0.027**
			Weight	41.4 ± 6.0	38.2 ± 3.6	36.6 ± 11.6	39.0 ± 3.1	3	1.170	0.332
		Nonelite	Height	146.9 ± 7.1	142. 7 ± 6.5	139.4 ± 5.7	139.9 ± 6.4	3	5.280	**0.002**
			Weight	41.9 ± 9.4	39.6 ± 7.1	36.7 ± 6.7	35.9 ± 6.9	3	2.355	0.080
	U14	Elite	Height	164.4 ± 9.4	162.4 ± 8.0	162.4 ± 13.0	163.7 ± 5.4	3	0.133	0.989
			Weight	49.4 ± 9.4	47.1 ± 7.3	49.9 ± 13.3	52.8 ± 4.9	3	0.284	0.837
		Nonelite	Height	157.9 ± 7.5	156.6 ± 8.8	156.5 ± 8.4	153.4 ± 9.7	3	0.976	0.409
			Weight	48.4 ± 6.8	45.5 ± 8.2	46.6 ± 7.9	44.4 ± 7.4	3	0.880	0.455
	U16	Elite	Height	174.1 ± 6.2	171.7 ± 6.7	174.0 ± 10.3	-	3	0.640	0.531
			Weight	63.3 ± 6.6	59.8 ± 5.7	64.8 ± 9.4	-	3	1.876	0.163
		Nonelite	Height	167.6 ± 6.9	168.7 ± 7.2	166.2 ± 7.5	168.3 ± 8.2	3	0.387	0.763
			Weight	58.0 ± 10.9	60.7 ± 9.4	57.3 ± 12.3	59.1 ± 11.9	3	0.283	0.837
Female	U12	Elite	Height	143.5 ± 6.9	145.8 ± 7.6	144.7 ± 7.1	147.5 ± 7.1	3	0.524	0.668
			Weight	36.0 ± 5.9	39.2 ± 10.0	40.0 ± 9.2	36.0 ± 6.1	3	0.735	0.538
		Nonelite	Height	146.3 ± 9.0	146.7 ± 12.0	137.2 ± 7.3	152.0 ± 12.4	3	1.919	0.167
			Weight	37.9 ± 10.1	42.9 ± 12.3	36.6 ± 9.2	51.4 ± 10.1	3	1.859	0.177
	U14	Elite	Height	156.72 ± 6.2	158.0 ± 7.4	155.4 ± 7.3	154.1 ± 4.9	3	0.577	0.633
			Weight	45.9 ± 4.6	48.6 ± 7.2	52.5 ± 9.1	46.0 ± 3.6	3	1.940	0.138
		Nonelite	Height	153.4 ± 4.5	153.4 ± 4.5	153.1 ± 6.0	150.6 ± 5.5	3	0.765	0.522
			Weight	47.5 ± 5.5	48.6 ± 4.8	47.4 ± 7.9	47.2 ± 6.0	3	0.087	0.967
	U16	Elite	Height	163.8 ± 4.7	161.3 ± 6.2	159.8 ± 5.4	160.6 ± 8.6	3	1.002	0.403
			Weight	56.9 ± 8.4	54.1 ± 7.8	54.0 ± 5.5	54.2 ± 5.9	3	0.461	0.711
		Nonelite	Height	162.5 ± 5.6	157.1 ± 3.0	157.7 ± 3.2	158.3 ± 5.6	3	2.232	0.105
			Weight	59.3 ± 7.1	50.9 ± 5.7	55.4 ± 4.8	53.3 ± 7.2	3	2.431	0.085

Note: BQ1 = first quartile; BQ2 = Second quartile; BQ3 = Third quartile; BQ4 = fourth quartile; n = number; % = percentage; x^2^ = chi-square value; gl = freedom degree; *p* = significance value.

**Table 4 ijerph-20-02015-t004:** Maturity status in the different elite and nonelite academies, differentiating between genders.

Category	Academy	Male			Female		
		Early	On Time	Late	Early	On Time	Late
U12	Elite	5 (10.4%)	40 (83.3%)	3 (6.3%)	3 (7.1%)	30 (71.4%)	9 (21.4%)
	Nonelite	6 (8.3%)	57 (79.2%)	9 (12.5%)	5 (25.0%)	14 (70.0%)	1 (5.0%)
U14	Elite	10 (25.6%)	23 (59.0%)	6 (1.,4%)	11 (24.4%)	28 (62.2%)	6 (13.3%)
	Nonelite	6 (7.2%)	63 (75.9%)	14 (16.9%)	2 (5.4%)	27 (73.0%)	8 (21.6%)
U16	Elite	15 (25.9%	43 (74.1%)	0 (0.0%)	9 (23.1%)	23 (59.0%)	7 (17.9%)
	Nonelite	5 (5.9%)	63 (74.1%)	17 (20.0%)	5 (14.7%)	23 (67.6%)	6 (17.6%)
TOTAL	Elite	30 (20.7%	106 (73.1%)	9 (6.2%)	23 (18.3%)	81 (64.3%)	22 (17.5%)
	Nonelite	17 (7.1%)	183 (76.3%)	40 (16.7%)	12 (13.2%)	64 (70.3%)	15 (16.5%)

**Table 5 ijerph-20-02015-t005:** Differences between elite and nonelite academies, taking into account category and gender.

**MALE**	**Under 12**				**Under 14**				**Under 16**			
	**Elite** **(n = 48)**	**Nonelite** **(n = 72)**			**Elite** **(n = 39)**	**Nonelite** **(n = 83)**			**Elite** **(n = 58)**	**Nonelite** **(n = 85)**		
**Variables**	**M ± SD**	**M ± SD**	**t**	** *p* **	**M ± SD**	**M ± SD**	**t**	** *p* **	**M ± SD**	**M ± SD**	**t**	** *p* **
Age	11.21 (±0.62)	10.72 (±0.56)	4.439	**0.000**	13.27 (±0.57)	12.77 (±0.57)	4.540	**0.000**	15.22 (±0.54)	14.72 (±0.53)	5.491	**0.000**
APHV (years)	13.35 (±0.59)	13.50 (±0.43)	−1.586	**0.115**	13.73 (±0.62)	13.99 (±0.50)	−2.357	**0.020**	13.75 (±0.56)	14.42 (±0.94)	−5.348	**0.000**
PHVD	−2.147 (±0.82)	−2.779 (±0.57)	4.982	**0.000**	-469 (±0.86)	-1.220 (±0.74)	4.977	**0.000**	1.747 (±0.69)	0.303 (±1.11)	7.745	**0.000**
Height (cm)	150.09 (±7.92)	142.55 (±7.05)	5.465	**0.000**	163.63 (±9.00)	156.03 (±8.69)	4.453	**0.000**	173.51 (±7.11)	1167.63 (±7.43)	4.725	**0.000**
Weight (kg)	39.21 (±7.56)	38.77 (±7.93)	0.310	0.757	49.01 (±8.93)	46.07 (±7.68)	1.867	0.064	62.62 (±7.05)	58.57 (±11.23)	2.650	**0.009**
BMI	17.79 (±1.50)	18.96 (±3.00)	−2.822	**0.006**	18.14 (±1.65)	18.86 (±2.22)	−1.786	0.077	20.78 (±1.68)	20.71 (±2.94)	0.190	0.850
**FEMALE**	**Under 12**				**Under 14**				**Under 16**			
	**Elite** **(n = 43)**	**Nonelite** **(n = 20)**			**Elite** **(n = 45)**	**Nonelite** **(n = 37)**			**Elite** **(n = 39)**	**No Elite** **(n = 34)**		
**Variables**	**M ± SD**	**M ± SD**	**t**	** *p* **	**M ± SD**	**M ± SD**	**t**	** *p* **	**M ± SD**	**M ± SD**	**t**	** *p* **
Age	11.00 (±0.58)	10.54 (±1.57)	1.272	0.217	12.82 (±0.57)	12.63 (±0.57)	1.476	0.144	14.97 (±0.78)	14.68 (±0.62)	1.748	0.085
APHV (years)	12.02 (±0.434)	11,66 (±0.566)	2.784	**0.007**	12.11 (±0.403)	12.241 (±0.364)	−1.528	0.130	12.763 (±0.534)	12.826 (±0.458)	−0.543	0.589
MO (years)	−1.025 (±0.677)	−1.124 (±1.351)	0.310	0.759	0.706 (±0.557)	0.389 (±0.550)	2.576	0.012	2.208 (±0.616)	1.853 (±0.451)	2.775	**0.007**
Height (cm)	144.90 (±7.03)	144.82 (±10.77)	0.034	0.973	156.64 (±6.66)	152.16 (±5.26)	3.323	0.001	161.74 (±5.83)	159.07 (±5.03)	2.083	**0.041**
Weight (kg)	37.73 (±7.83)	41.45 (±11.14)	−1.345	0.189	48.32 (±6.88)	47.58 (±6.06)	0.512	0.610	55.11 (±7.23)	54.97 (±6.85)	0.083	0.934
BMI	17.86 (±2.77)	19.42 (±3.26)	−1.936	0.054	19.69 (±2.45)	20.48 (±1.78)	−1.703	0.093	21.06 (±2.54)	21.69 (±2.22)	−1.132	0.262

Note: M = mean; SD = standard deviation; *p* = significance value; t = T student; APHV = age of peak height velocity; MO = maturity offset; BMI = body mass index.

**Table 6 ijerph-20-02015-t006:** Correlation between coach’s assessment of player performance and player anthropometry, in different age categories by type of academy and gender.

Gender	Category	Type of Academy	Height		Weight		APHV	
			R-Pearson		R-Pearson		R-Pearson	
			Cp	Fe	Cp	Fe	Cp	Fe
Male	U12	Elite	0.139	−0.075	−0.147	−0.236	−0.202	−0.053
		Nonelite	0.375 **	0.247	0.144	0.036	−0.223	−0.214
	U14	Elite	0.049	−0.058	0.021	−0.062	0.090	0.066
		Nonelite	0.135	0.030	0.113	−0.004	−0.133	−0.110
	U16	Elite	−0.022	0.112	0.061	0.046	−0.139	−0.279 *
		Nonelite	0.334 **	0.355 **	0.181	0.211	−0.065	−0.177
	Total	Elite	-	-	-	-	−0.170 *	−0.173 *
		Nonelite	-	-	-	-	−0.071	−0.164 *
Female	U12	Elite	0.074	−0.086	0.028	0.013	0.024	−0.008
		Nonelite	0.133	−0.180	0.154	−0.158	0.181	0.059
	U14	Elite	−00.092	−0.100	−0.256	−0.248	0.259	0.125
		Nonelite	0.011	0.208	0.268	0.233	−0.013	−0.166
	U16	Elite	−0.182	−0.257	−0.111	−0.136	0.018	0.182
		Nonelite	0.363 **	0.056	0.182	−0.047	−0.181	−0.032
	Total	Elite	-	-	-	-	0.052	0.019
		Nonelite	-	-	-	-	−0.176	−0.063

Note: U = under; Cp = coach’s assessment of current performance; Fe = coach’s assessment of future expectation of success, * = *p* < 0.05; ** = *p* < 0.01.

**Table 7 ijerph-20-02015-t007:** Analysis of the coaches’ current and future expectation of success and the RAE.

Gender	Category	Academy		Quarter				GL	F	*p*
				BQ1	BQ2	BQ3	BQ4			
Male	U12	Elite	Cp	3.67	3.76	3.72	3.6	3	0.052	0.984
			Fe	3.92	3.85	3.69	3.17	3	0.886	0.456
		Nonelite	Cp	3.42	3.02	2.77	3.0	3	1.743	0.167
			Fe	3.67	3.57	3.43	3.41	3	0.438	0.727
	U14	Elite	Cp	3.24	3.25	3.43	2.5	3	0.567	0.640
			Fe	3.39	3.40	3.31	2.67	3	0.446	0.722
		Nonelite	Cp	3.62	3.12	3.14	3.28	3	1.216	0.309
			Fe	3.81	3.32	3.62	3.38	3	1.508	0.219
	U16	Elite	Cp	2.94	3.48	3.47	-	2	3.349	0.042
			Fe	3.08	3.42	3.60	-	2	2.208	0.120
		Nonelite	Cp	3.04	3.08	3.43	3.20	3	0.916	0.437
			Fe	3.18	3.31	3.62	3.17	3	1.144	0.336
Female	U12	Elite	Cp	3.18	3.60	3.60	2.50	3	2.127	0.113
			Fe	3.81	3.80	4.10	2.67	3	3.473	**0.025**
		No Elite	Cp	4.20	3.20	3.67	4.00	3	1.162	0.355
			Fe	4.00	3.00	3.50	3.75	3	0.594	0.628
	U14	Elite	Cp	3.77	3.56	2.44	3.43	3	2.516	0.072
			Fe	4.09	4.25	3.00	4.14	3	3.591	**0.021**
		No Elite	Cp	2.83	3.36	3.22	3.37	3	0.733	0.540
			Fe	3.58	3.29	3.39	3.70	3	0.412	0.745
	U16	Elite	Cp	3.07	3.50	3.11	2.90	3	0.958	0.423
			Fe	3.46	3.86	3.56	3.40	3	0.511	0.677
		Nonelite	Cp	3.33	3.08	3.31	3.04	3	0.379	0.769
			Fe	3.22	3.83	3.00	3.72	3	1.853	0.159

Note: U = under; Cp = coach’s assessment of current performance; Fe = coach’s assessment of future expectation of success; gl = freedom degree; *p* = significance value.

## Data Availability

Not applicable.

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
