# Peer review of "Age and Maturation Matter in Youth Elite Soccer, but Depending on Competitive Level and Gender"

_ijerph, 2023, doi:10.3390/ijerph20032015_

Round 1

Reviewer 1 Report

I congratulate the authors for conducting the study.

What are the practical applications of the study?

How can trainers guide training sessions?

What differences should be considered between elite and non-elite athletes?

How to apply the measures in professional practice?

Authors could insert an excel spreadsheet for other trainers to download the material. How to interpret the findings?

Author Response

Reviewers' Comments to Author

Reviewer 1

On behalf of the authors, I would like to thank to the Reviewer 1 his/her positive feedback and constructive comments and suggestions on our original submission.

We apologize to you for the language mistakes. The manuscript has been revised again by a native English speaker with wide experience in translations for the field of Sport Sciences. We undoubtedly believe that the manuscript has been improved.

We have detailed our responses to the Reviewer 1 comments and suggestions on the pages that follow.

---

R#1.1: I congratulate the authors for conducting the study.

  1. We are glad and grateful for your congratulations. We have intented to improve our manuscript via your meaningful contributions.

---

R#1.2: What are the practical applications of the study?

  1. According to the reviewer recommendation we have added a new section of practical applications (4.3. Practical Applications, page 16, lines 528-547).

We have described how our findings could be useful in improving the development of youth soccer. We have added ideas on 3 main points: a) Longitudinal monitoring of the players’ maturity status providing elementary information about individual / team Maturity offset, PHV, and RAE; b) Individualization of training process according the players´ maturity state, and ; c)  Combining coaches´ and performance assesment with objective information about players  maturity status provided by sports scientist.

---

R#1.3. : How can trainers guide training sessions?

  1. This interesting question has been addressed in the above mentioned section "Practical application" (page 16, lines 528-547). We have highlighted the relevance of implementing a continuous control of the maturity status, in order to be able to deal more individually with the different physical loads and musculo-skeletal demands.

One of the methods that seems to be giving positive results at present is Bio-Banding, where players are grouped by biological age instead of birthdate, to mitigate the physical advantages of the more developed players, creating a more competitive climate between equals. This method can be implemented in different ways: bio-banding within the team in some tasks of the training session, mixing together players from different teams of the club for some training sessions of the week, attending or organising Bio-Banded tournaments...

---

R#1.4: What differences should be considered between elite and non-elite athletes?

  1. Different studies have shown that in the formative periods, the maturational processes affect performance, and therefore, a higher performance can be seen in those players who are physically more mature and developed. It is usual, as showed in our study, that elite players are more advanced in terms of maturity than their non-elite counterparts. In addition, elite players are in a more favourable context, with more training days, better facilities and resources and more educated coaches.

In the non-elite context,  the talent selection processes are less demanding, and more usual to find more late developed players than in the elite academies. In these context of practice, with less educated coaches it is also likely that they have never been tauhght about the influence of relative age and maturity in the sporting environment . It could be very interesting to improve coaches education about the maturation effects on players development to help them implement adaptations in the training procceses to better integrate late maturing players .

---

R#1.5: How to apply the measures in professional practice?

In our opinion, this interesting question has been replied in the R#1.2, R#1.3  and R#1.4, and with the inclusion in the paper of the section "Practical application" (4.3. Practical Applications, page 16, lines 528-547). Regarding the application in professional practice, we have suggested some ideas related with assessment and control of  the state of maturity that could be implemented in different departments (Methodology, Strength and Conditioning, Psychology, Medical Department,…) of the clubs. These include some recommendations for intervention at different stages: a) Interdisciplinary discussion in the process of detection, selection and promotion of talented players;  b) Longitudinal monitoring of the individual / team state of maturity based on maturity offset, PHV, and RAE to make decisions, and c) Individualization of training process according the player´s state of maturity.

---

R#1.6: Authors could insert an excel spreadsheet for other trainers to download the material. How to interpret the findings?

We agree with the reviewer that it is definitely a good idea to try help other researchers or coaches interested in the topic by sharing this material. Therefore, we attach the two Excel files used in the research, so that it can be downloaded by other colleagues.

---

Reviewer 2 Report

Dear Authors.

The following is a review of the article entitled "Age and Maturation Matter in Youth Elite Soccer. Coaches´ Assessment of Players’ Performance is Modulated by Maturity Status and Relative Age, Depending on Gender and Competitive Level”, which the aimed to explore the relevance of relative age effect, maturity status and anthropometry, and their influence on coaches’ assessment of players´ performance, analyzing both genders and 16 different types of academies (elite vs. non-elite).

Thank you very much for thinking of me as a reviewer for this study.

After carefully reading the manuscript, I set forth comments and suggestions for the authors:

Title: Incorrect. It should be simplified. No dot should appear in the middle.

Abstract: Correct. Do not use acronyms in the abstract.

Keywords: Correct.  

Introduction: Correct. It is necessary to add a hypothesis for this study.

Materials and Methods: Sample: Explain the differences between the elite and non-elite group.  

Coaches’ assessment of current and future player’s success. What questions were asked to assess the coaches' assessment of the players' current and future success? Are the questions in the questionnaire validated by any recent research?

This method is unscientific. It should be improved.

Results: Good presentation of the results. Figures 1 and 2 are irrelevant. Add these results inside figure 3 and 4.

Discussion: A good discussion. Good practical application.

Conclusions: Extend the conclusions to answer all the objectives of the study.

References: Corrects but some errors found in the references (7, 59, 67). Please check them.

Regards.

Author Response

Reviewers' Comments to Author

Reviewer 2

On behalf of the authors, I would like to thank to the Reviewer 2 for his/her constructive comments and suggestions, which have enabled us to develop a more comprehensive work.

We have detailed our responses to comments and suggestions on the following pages (and Please see the attachment, including the Cover Letter, response to Reviewer 1, and at the end, the point-by-point response to the reviewer’s 2 comments  )

---

R#2.1: Authors suggestions

  1. It is a pleasure for us to receive your point of view about this topic, which will undoubtedly allow us to improve our knowledge of the subject and improve the quality of our manuscript.

In the following, we will try to respond to the reviewer's various comments and suggestions.

---

R#.2.2. Title: Incorrect. It should be simplified. No dot should appear in the middle.

  1. Thank you for your suggestion. In our first attempt to define the title of the study, we tried to include the key words that would define our main objectives. However, as the reviewer properly points out, the large number of variables involved in this research made it difficult to come up with a shorter title. In this new version of the manuscript we have reduced the title to “Age and Maturation Matter in Youth Elite Soccer, but depending on Competitive Level and Gender”.

---

R#2.3. Abstract: Correct. Do not use acronyms in the abstract.

  1. Thanks. Acronyms have been corrected. We have tried to follow the recommendation for writing scientific papers trying to avoid acronyms in the abstract. However, the use of acronism or initialism is allowed in those cases where the acronym is commonly understood and used multiple times in the abstract (it is our case due the name of main dependent and independent variables). This is the reason because we have maintained the use of acronyms in the abstract, but all of them have been spelled out (defined) in the abstract, and then spelled out again the first time that we used in the body of the paper. This was done with the acronyms: RAE (Relative Age Effect), the age categories (U12, U14 and U16, from Under 12, Under 14 and Under 16 respectively), maturity offset (MO) and Peak Height Velocity (PHV).

---

R#2.3. Introduction: Correct. It is necessary to add a hypothesis for this study.

  1. The reviewer raises a very interesting point here, because we set the objectives only. According to the reviewer recommendation, we have added some paragraphs including our hypothesis based on the literature analyzed at the end of the introduction section (page 3, lines 106-113).

---

R#2.4. Materials and Methods: Sample: Explain the differences between the elite and non-elite group.  

  1. We appreciate the reviewer's suggestion. In the previous version of our manuscript, we included in the Introduction section (pages 1-2, lines 38-44), a characterisation of the differences between elite and non-elite (or social) soccer academies according to Merce's (2006) definition. In the revised version of our manuscript ,we have kept this definition in the introduction (page 1-2, lines 39-45) to contextualise the study. However, in order to clarify one of the key independent variables in our study, in the Method section (page 3, lines 120-123) we have more clearly defined the criteria that define elite and non-elite academies in our research.

---

R#2.5. Coaches’ assessment of current and future player’s success. What questions were asked to assess the coaches' assessment of the players' current and future success? Are the questions in the questionnaire validated by any recent research? This method is unscientific. It should be improved.

  1. Thank you for your interesting reflections on these methodological aspects related to the subjective assessment of current performance levels and future expectations of player´s success by coaches.

Previous studies included in the literature reviewed in the present study justify the interest of including direct questions addressed to coaches to assess their perception of performance, effectiveness or individual performance of players or the team (see citations 61 and 62 in the references section).

In our research we included several direct questions to coaches, which aimed to obtain interesting information about their perceptions of player´s current performance, as well as their expectations of their future potential, since we consider that these may be key to understanding their behaviour (conscious or unconscious) in the coaching processes (selection and promotion of players).

In order to clarify the content of these questions, and following the reviewer's indications, we have included the statement of the questions in section 2.2.4 of the Material and Methods section (page 4-5, lines 178-183).

---

R#2.6. Results: Good presentation of the results. Figures 1 and 2 are irrelevant. Add these results inside figure 3 and 4.

R: Thanks for you suggestion, an attempt has been made to integrate figures 1 and 2 into figures 3 and 4, but this has not been possible due to the format and size of the graph, and it is not clear what is intented to be reflected.

Therefore, we propose the scenario of linking figure 1 and 2, highlighting the differences in RAE by academy and gender (see new Figure 1, page 6, line 234).

On the other hand, we have combined figures 3 and 4, highlighting the differences in maturity status within each birth quarter, by academy and gender (see new Figure 2, page 7, line 271).

---

R#2.7. Discussion: A good discussion. Good practical application.

  1. We appreciate the reviewer´s positive feedback on the "Discussion" and "Practical Application" sections. This latter has been incorporated as an addition to the new corrected version of the manuscript.

---

R#2.8. Conclusions: Extend the conclusions to answer all the objectives of the study.

  1. To address the suggestion, we have added a section on conclusions (4.2 Conclusions, pages 15-16, lines 511-525), extending also in the section on 4.3 Practical applications (page16, lines 528-547).

---

R#2.9. References: Corrects but some errors found in the references (7, 59, 67).

  1. Thank you for your detailed and thorough reading of the references section. The references indicated have been corrected.

We thank Reviewer 2 for very detailed comments and suggestions to improve the quality of our previous submission. We hope that you will now find the enclosed manuscript suitable for publication in International Journal of Environmental Research and Public Health.

Yours sincerely,

Florentino Huertas on behalf of all authors

Round 2

Reviewer 2 Report

Dear Authors.

After the first revision of the article entitled "Age and Maturation Matter in Youth Elite Soccer, but depend-2 ing on Competitive Level and Gender" the authors have improved the points indicated. The manuscript is considered ready for publication.

Regards